# Diketopiperazine and Diphenylether Derivatives from Marine Algae-Derived *Aspergillus versicolor* OUCMDZ-2738 by Epigenetic Activation

**DOI:** 10.3390/md17010006

**Published:** 2018-12-22

**Authors:** Wen Liu, Liping Wang, Bin Wang, Yanchao Xu, Guoliang Zhu, Mengmeng Lan, Weiming Zhu, Kunlai Sun

**Affiliations:** 1Zhejiang Provincial Engineering Technology Research Center of Marine Biomedical Products, School of Food and Pharmacy, School of Marine Science and Technology, Zhejiang Ocean University, Zhoushan 316022, China; yxlliuwen@outlook.com (W.L.); wangbin4159@hotmail.com (B.W.); 2State Key Laboratory of Functions and Applications of Medicinal Plants, Guizhou Medical University, Guiyang 550014, China; lipingw2006@163.com (L.W.); m18586818694@163.com (Y.X.); 3Key Laboratory of Marine Drugs, Ministry of Education of China, School of Medicine and Pharmacy, Ocean University of China, Qingdao 266003, China; zhuguoliang@ecust.edu.cn (G.Z.); lanmengmeng90@163.com (M.L.); 4State Key Laboratory of Bioreactor Engineering, East China University of Science and Technology, Shanghai 200237, China; 5Open Studio for Druggability Research of Marine Natural Products, Pilot National Laboratory for Marine Science and Technology (Qingdao), Qingdao, 266237, China

**Keywords:** chemical-epigenetic method, endophytic fungus, *Aspergillus versicolor*, enantiomer, antimicrobial activity

## Abstract

A chemical-epigenetic method was used to enhance the chemodiversity of a marine algicolous fungus. Apart from thirteen known compounds, (+)-brevianamide R ((+)-**3**), (‒)-brevianamide R ((‒)-**3**), (+)-brevianamide Q ((+)-**4**), (‒)-brevianamide Q ((‒)-**4**), brevianamide V ((+)-**5**), brevianamide W ((‒)-**5**), brevianamide K (**6**), diorcinol B (**7**), diorcinol C (**8**), diorcinol E (**9**), diorcinol J (**10**), diorcinol (**11**), 4-methoxycarbonyldiorcinol (**12**), two new compounds, (+)- and (‒)-brevianamide X ((+)- and (‒)- **2**)), as well as a new naturally occurring one, 3-[6-(2-methylpropyl)-2-oxo-1*H*-pyrazin-3-yl]propanamide (**1**), were isolated from chemical-epigenetic cultures of *Aspergillus versicolor* OUCMDZ-2738 with 10 µM vorinostat (SAHA). Compared to cultures in the same medium without SAHA, compounds **1**–**4**, **8**, **9**, **11**, and **12** were solely observed under SAHA condition. The structures of these compounds were elucidated based on spectroscopic analysis, specific rotation analysis, ECD, and X-ray crystallographic analysis. (±)-**3**, (±)-**4**, and (±)-**5** were further resolved into the corresponding optically pure enantiomers and their absolute configurations were determined for the first time. Compounds **11** and **12** showed selective antibacterial against *Pseudomonas aeruginosa* with a minimum inhibitory concentration (MIC) of 17.4 and 13.9 μM, respectively. Compound **10** exhibited better α-glucosidase inhibitory activity than the assay control acarbose with IC_50_ values of 117.3 and 255.3 μM, respectively.

## 1. Introduction

Marine natural products (NPs) are an important source for developing new drugs, especially those from marine-derived fungi which possess novel structures and significant biological activities [1,2]. *Aspergillus* is a kind of productive fungal genus and has been reported to produce many new NPs [3], such as alkaloids [4,5,6], diketopiperazines [7,8], xanthones [8], and diphenyl ethers [9]. However, the whole genome sequencing studies of fungi have indicated that most genes of fungi were silent and not expressed in the conventional experimental conditions. Fungi treated with DNA methyltransferase and histone deacetylase inhibitors enhanced NPs’ chemical diversity, demonstrating that small molecule epigenetic modifiers are effective tools for rationally generating new and bioactive NPs [10]. Both molecular-based and chemical approaches targeting histone and DNA posttranslational processes have the great potential for rationally activating or suppressing NP-encoding gene clusters [11,12,13]. Epigenetic modification has become a widely applied strategy to find new and bioactive NPs today [14,15,16].

In our continuous study on the discovery of new and bioactive NPs from algicolous microorganisms isolated from *Enteromorpha prolifera* by epigenetic modification [12,13] and other strategies [17,18,19], we found that the algicolous *Aspergillus versicolor* OUCMDZ-2738 could produce different NPs after chemical-epigenetic modification by 10 µM SAHA. From the epigenetically modified fermentation of *A. versicolor* OUCMDZ-2738, we isolated and identified a new naturally occurring compound **1**, two new enantiomers ((+)-**2** and (‒)-**2**), and thirteen known compounds (**3**–**12**). Their structures were determined as 3-[6-(2-methylpropyl)-2-oxo-1*H*-pyrazin-3-yl] propanamide (**1**), (+)-brevianamide X ((+)-**2**) and (‒)-brevianamide X ((‒)-**2**), (±)-brevianamide R (**3**) [20], (±)-brevianamide Q (**4**) [20], brevianamide V ((+)-**5**) [21], brevianamide W ((‒)-**5**) [22], brevianamide K (**6**) [23], diorcinol B (**7**) [24], diorcinol C (**8**) [24], diorcinol E (**9**) [24], diorcinol J (**10**) [24], diorcinol (**11**) [25], and methyl diorcinol-4-carboxylate (**12**) [26,27], respectively (Figure 1). (±)-Brevianamide R (**3**) and (±)-brevianamide Q (**4**) were further resolved as (+)- and (‒)- brevianamide R ((+)- and (‒)- **3**) and (+)- and (‒)- brevianamide Q ((+)- and (‒)- **4**), respectively.

## 2. Results and Discussion

Compound **1** was isolated as a brown, amorphous powder. The molecular formula was determined as C_11_H_17_O_2_N_3_ by a HRESIMS peak at *m*/*z* 246.1205 [M + Na]^+^ (calcd for C_11_H_17_O_2_N_3_Na, 246.1213) (Appendix A). 1D (Table 1 and Appendix A) and HSQC (Appendix A) NMR spectra displayed one sp^2^ methine (δ_C_ 120.8, δ_H_ 7.01), one sp^3^ methine (δ_C_ 27.5, δ_H_ 1.91), three sp^3^ methylenes (δ_C_ 27.6, δ_H_ 2.79; δ_C_ 31.1, δ_H_ 2.41; δ_C_ 38.4, δ_H_ 2.25), and two methyls (δ_C_ 21.9, δ_H_ 0.86; δ_C_ 21.9, δ_H_ 0.84) (Table 1). The HMBC (Appendix A and Figure 2) correlations from H-7 (δ_H_ 2.79), H-8 (δ_H_ 4.64), and NH_2_ (δ_H_ 7.30, δ_H_ 6.72) to carbonyl carbon (δ_C-9_ 173.7) along with the ^1^H–^1^H COSY (Appendix A and Figure 2) between H-7 and H-8 indicated a 3-substituted propionamide moiety. ^1^H–^1^H COSY correlations from H-10 (δ_H_ 2.25) to H-12 (δ_H_ 0.86), and H-13 (δ_H_ 0.84) through H-11 (δ_H_ 1.91), demonstrated an isobutyl fragment. The remaining fragment, C_4_H_2_ON_2_, could be a pyrazin-2(1*H*)-one nucleus from the HMBC correlations from H-4 (δ_H_ 7.01) to C-3 (δ_C_ 155.2) and C-6 (δ_C_ 138.0). Furthermore, the key HMBC correlations from H-10 to C-5 (δ_C_ 120.8) and C-6, and H-7 to C-2 (δ_C_ 156.2) and C-3, allowed for the linkage of the isobutyl to C-6 and the 3-substituted propionamide moiety to C-3, respectively. Thus, structure **1** was established as 3-[6-(2-methylpropyl)-2-oxo-1*H*-pyrazin-3-yl] propenamide, which was only recorded in the Aurora Fine Chemicals with ID number K13.052.726 without any physical and chemical properties reported. Consequently, **1** was identified as a new naturally occurring compound.

The racemic **2** was isolated as light yellow crystal with the molecular formula of C_21_H_23_N_3_O_4_ from a HRESIMS peak at *m*/*z* 404.1570 [M + Na]^+^ (calcd for C_21_H_23_N_3_O_4_Na, 404.1581) (Appendix A). Its 1D (Table 1 and Appendix A) and HSQC (Appendix A) NMR spectra displayed two methyls (δ_C_ 28.1, δ_H_ 1.57; δ_C_ 28.3, δ_H_ 1.54), two sp^3^ methylenes (δ_C_ 44.7, δ_H_ 3.96, δ_H_ 3.72; δ_C_ 29.0, δ_H_ 1.93, δ_H_ 2.41), one sp^3^ methine (δ_C_ 76.4, δ_H_ 4.41), five sp^2^ methines (δ_C_ 115.1, δ_H_ 7.29; δ_C_ 120.2, δ_H_ 7.37; δ_C_ 121.3, δ_H_ 7.07; δ_C_ 122.6, δ_H_ 7.12; δ_C_ 122.6, δ_H_ 7.43), and one sp^2^ methylene (δ_C_ 112.6, δ_H_ 5.10, 5.13), which demonstrated the presence of a monosubstituted double bond (Table 1). ^1^H–^1^H COSY (Appendix A and Figure 2) from H-21 (δ_H_ 6.11) to H-22 (δ_H_ 5.10; δ_H_ 5.13) and the HMBC (Appendix A and Figure 2) from H-21 (δ_H_ 6.11) to C-20 (δ_C_ 40.5), C-23 (δ_C_ 28.1), and C-24 (δ_C_ 28.3), along with H-23 (δ_H_ 1.57) to C-20 and C-24, which demonstrated the presence of a 2-methylbut-3-en-2-yl fragment. The ^1^H–^1^H COSY correlations from H-13 (δ_H_ 7.37) to H-16 (δ_H_ 7.43), along with the HMBC from H-13 (δ_H_ 7.37) to C-12 (δ_C_ 127.4), C-11 (δ_C_ 104.5), and C-17 (δ_C_ 127.4), and H-16 (δ_H_ 7.43) to C-12 and C-17, which demonstrated the presence of a 2,3-bisubstituted indole ring. Moreover, the key HMBC from H-21, H-23, and H-24 (δ_H_ 1.54) to C-19 (δ_C_ 146.2) connected the 2-methylbut-3-en-2-yl group to C-19 of the indole ring. Chemical shifts of C-1 (δ_C_ 165.7) and C-4 (δ_C_ 161.3) indicated two amide carbonyl carbons. In addition, ^1^H–^1^H COSY correlations from H-6 (δ_H_ 3.72; δ_H_ 3.96) to H-8 (δ_H_ 4.41) through H-7 (δ_H_ 1.93; δ_H_ 2.41) and HMBC correlations from H-6 to C-4 and C-9 (δ_C_ 91.0), H-7 to C-9, H-8 (δ_H_ 4.41) to C-1 and C-9 and, H-10 (δ_H_ 7.29) to C-3 (δ_C_ 125.1) and C-4 were observed. This data indicated a 2-methylene-(2,3-dihydroxytetrahydro) pyrrolo[1,2-*a*] diketopiperazine nucleus. The key HMBC correlations from H-10 to C-11, C-12, and C-19 (Figure 2) allowed for the linkage of the isopentenylindole and tetrahydropyrrolo[1,2-*a*] diketopiperazine moieties by a C-11–C-10 sigma bond. Thus, the planar structure of **2** was determined as 9-hydroxybrevianamide Q. A literature survey showed that **2** has the same planar structure as brevianamide U without resolution of the configuration (${\left[\alpha \right]}_{D}^{21}$ +20 (*c* 0.17, MeOH)) [21]. Although their ^1^H and ^13^C NMR data were similar, they displayed obvious different chemical shifts of C-1 and C-9 in ^13^C NMR, indicating the different configurations. The configuration of **2** was confirmed by X-ray single crystallographic diffraction that clearly indicated *trans*-configuration between 8-OH and 9-OH and the *Z*-geometry of the ∆^3(10)^-double bond (Figure 3). However, the X-ray crystallographic results were space group P-1 and the planus line of the electronic circular dichroism (ECD) spectrum indicated **2** might be a pair of enantiomers. Thus, two enantiomers with 1.1:1 ratio of HPLC peaks area were separated by a chiral column (Appendix A), whose specific rotations were ${\left[\alpha \right]}_{D}^{25}$ +44 and –40 (*c* 0.05, MeOH), respectively. The absolute configurations of (+)-**2** and (–)-**2** were respectively assigned as (8*S*, 9*S*) and (8*R*, 9*R*) by comparing the experimental and calculated ECD values obtained using TD-DFT at the B3LYP/6-31G(d) level [19,28,29] (Figure 4). Therefore, we named (+)-**2** and (–)-**2** as (+)-brevianamide X and (–)-brevianamide X, respectively.

The planar structures of compounds **3**–**5** were reported previously without assignment of the absolute configurations [20,22]. Their small specific rotations, ${\left[\alpha \right]}_{D}^{25}$ +10 (*c* 0.82, MeOH) for **3**, ${\left[\alpha \right]}_{D}^{25}$ –7 (*c* 1.75, MeOH) for **4**, and ${\left[\alpha \right]}_{D}^{25}$ +19 (*c* 0.33, MeOH) for **5**, indicated that compounds **3**–**5** might be enantiomeric mixtures, perhaps having an excess of one of the enantiomers. They were analyzed and resolved by chiral HPLC. As we expected, (±)-**3**, (±)-**4**, and (±)-**5** were further separated into their optically pure enantiomers (Appendix A). The absolute configurations were determined by comparing the experimental ECD with those calculated using TD-DFT at the B3LYP/6-31G(d) level (Figure 4). As a result, compounds (+)-**3** and **4** were assigned to have a 9*R*-configuration, while the (–)-**3** and **4** were assigned a 9*S*-configuration. This is the first time that racemic brevianamides R (**3**), Q (**4**), and V (**5**) have been resolved into their chiral enantiomers and their absolute configurations have been determined.

The structures of **6**–**11** were determined by comparing their NMR data and specific rotations those reported [23,24,25]. Although the structure of **12** has been previously reported in the references, there were no ^13^C NMR data [26] and nor NMR solvent reported [27], and chemical shifts of **12** (Table 2) were obviously different to those in ref. [27]. The difficulty was in the assignment of the position of the methoxycarbonyl group at C-2′ or C-4′. Thus, its dimethyl ether (**12a**) was prepared from **12** by reaction with MeI/K_2_CO_3_ and the 2D NMR of both **12** (Appendix A) and **12a** (Appendix A) were recorded. The ROESY correlations of CH_3_O-5′ (δ_H_ 3.72) with H-6′(δ_H_ 6.61) and the protons (δ_H_ 3.78) of the methyl ester in **12a** (Appendix A and Figure 5) located the methoxycarbonyl group at C-4′. Therefore, the structures of compounds **12** and **12a** were determined as methyl diorcinol-4-carboxylate and methyl 3,3′-di(*O*-methyl) diorcinol-4-carboxylate, respectively.

A plausible biosynthetic relationship linking **2**–**6** through a sequence of oxidative transformations and hydrolysis or methanolysis is illustrated in Figure 6. Positions 8 and 9 in **6** have no steric hindrance when performing electrophilic addition or oxidation. Therefore, it is easy to form enantiomeric mixtures or excess of **2**–**5**.

HPLC-UV was used to identify which compounds are solely produced under SAHA. Both extracts, generated with and without addition of SAHA, were analyzed using the same elution gradient. Compared to those in the same medium without SAHA, epigenetic modification with SAHA enhanced the NP diversity of *A. versicolor* OUCMDZ-2738 and the compounds **1**–**4**, **8**, **9**, **11**, and **12** were solely observed under SAHA condition. This demonstrated SAHA as the actual inducer to produce compounds **1**–**4**, **8**, **9**, **11**, and **12** (Appendix A and Figure 7), further indicating that chemically epigenetic regulation is an effective method for expanding the chemical space of fungal natural products.

All the isolated compounds (**1**–**12**) were evaluated for antibacterial activity against a panel of pathogenic microorganisms, including four Gram-positive bacteria, (*Bacillus subtilis* ATCC6051, *Clostridium perfringens* ATCC13048, *Staphylococcus aureus* ATCC6538, and *Staphylococcus aureus* ATCC25923), two Gram-negative bacteria (*Pseudomonas aeruginosa* ATCC10145 and *Escherichia coli* ATCC11775), and two pathogenic fungi (*Candida albicans* ATCC10231, *Candida glabrata* ATCC2001). The results showed that compound **11** selectively inhibited *P. aeruginosa* while **12** inhibited *P. aeruginosa* and *C. glabrata* with MIC values of 17.4, 13.9, and 27.8 μM, respectively (Table 3). The inhibitory activities of **1**–**12** against *α*-glucosidase from *Saccharomyces cerevisiae* were also evaluated using the pNPG method. Only compounds **9**–**11** displayed *α*-glucosidase inhibitory activity with IC_50_ values of 117.3, 237.4 and 275.3 μM, respectively. Acarbose, used as the assay control, had an IC_50_ of 255.3 μM.

## 3. Materials and Methods

### 3.1. General Experimental Procedures

Optical rotations were measured using an AUTOPOL1 polarimeter (Rudolph Research Analytical, Hackettstown, NJ, USA). UV spectra were measured on a Beckman DU 640 spectrophotometer (Beckman Coulter, Inc., Brea, CA, USA). ECD spectra were measured on JASCOJ-715 spectropolarimeter (JASCO Corporation, Tokyo, Japan). IR spectra were taken on a Nicolet Nexus 470 spectrophotometer (Thermo Nicolet Corporation, Madison, WI, USA) as KBr disks. 1D (one-dimensional) and 2D (two-dimensional) NMR spectra were recorded on an INOVA-400MHz spectrometer with TMS (tetramethylsilane) as the internal standard. ESIMS and HRESIMS were carried out on a Waters 2695 LCQ-MS liquid chromatography mass spectrometer (Thermo Finnigan, San Francisco, CA, USA). Semi-preparative HPLC separations were performed using a Hitachi Primaide Organizer HPLC system (Hitachi High-Tech Science Corporation, Tokyo, Japan). YMC-Pack ODS-A column (S-5 μm, 12 nm, 250 × 4.6 mm and S-5 μm, 12 nm, 250 × 10 mm, Shenzhen Chemist Technology Co. Ltd., Shenzhen, China). Chiral column (Lux^®^ 5 µm Cellulose-2 and Lux^®^ 5 µm i-Cellulose-5, LC Column 250 × 4.6 mm., Phenomenex, CA, USA). Thin layer chromatography (TLC) and column chromatography (CC) were performed on plates precoated with silica gel GF 254 (10–40 µm) and over silica gel (200–300 mesh, Qingdao Marine Chemical Factory, Qingdao, China), respectively. Chemical reagents for isolation were analytical grade (Tianjin Kaixin Chemical Industry Co., Ltd., Tianjin, China); chromatographic grade MeOH for HPLC was from Tianjin Siyou Fine Chemicals Co., Ltd., Tianjin, China.

### 3.2. Fungal Material

The fungus *Aspergillus versicolor* OUCMDZ-2738 was isolated from a piece of fresh tissue from the *Enteromorpha prolifera*, collected from the Shilaoren beach, Qingdao, China, in July 2012 [12,13]. The pure cultures were deposited in the Key Laboratory of Marine Drugs, School of Medicine and Pharmacy, Ocean University of China, Qingdao, Shandong province, China, with the GenBank (NCBI) accession number MH150818.

### 3.3. Cultivation and Extraction of Strain OUCMDZ-2738

The cultivation and extraction process were the same described in previous papers [12,13], and the EtOAc solutions were concentrated under reduced pressure to give a crude extract (13.1 g).

### 3.4. Purification

The crude extract was applied to a silica gel (200–300 mesh) column and was separated into twelve fractions (Fr.1 to Fr.12) using a step-gradient elution of petroleum ether/EtOAc (0–100%) and CHCl_3_/MeOH.

Fr.1 (0.2 g) was purified by semipreparative HPLC (50:50 MeOH/H_2_O, 4 mL/min) to give **1** (11.6 mg, *t*_R_ 9.6 min). Fr.12 (0.24 g) was purified by semipreparative HPLC (70:30 MeOH/H_2_O, 4 mL/min) to give the racemic-**2** (8.0 mg, *t*_R_ 6.5 min). Fr.7 (0.85 g) was separated on a preparative TLC eluted with CH_2_Cl_2_/CH_3_OH (*v/v*, 30:1), then subfraction Fr.7-1 (142.0 mg) was purified by semipreparative HPLC (70:30 MeOH/H_2_O, 4 mL/min) to give compound **6** (78.6 mg, *t*_R_ 6.4 min). Subfraction Fr.7-3 (101 mg) was purified by semipreparative HPLC (50:50 MeOH/H_2_O, 4 mL/min) to give racemic-**4** (9.3 mg, *t*_R_ 8.3 min). Subfraction Fr.7-4 (75 mg) was purified by semipreparative HPLC (60:40 MeOH/H_2_O, 4 mL/min) to give **9** (10.5 mg, *t*_R_ 7.0 min) and **8** (5.3 mg, *t*_R_ 8.1 min). Fr.8 (0.42 g) was separated on a preparative TLC eluted with CH_2_Cl_2_/MeOH (*v*/*v*, 50:1), then subfraction Fr.8-1 (178.3 mg) was purified by semipreparative HPLC (65:35 MeOH/H_2_O, 4 mL/min) to give racemic-**3** (7.3 mg, *t*_R_ 8.0 min) and racemic-**5** (90.0 mg, *t*_R_ 8.5 min). Fr.5 (0.21 g) was purified by semipreparative HPLC (70:30 MeOH/H_2_O, 4 mL/min) to give **7** (40.8 mg, *t*_R_ 9.7 min) and **10** (9.8 mg, *t*_R_ 8.7 min). Fr.4 (0.12 g) was purified by semipreparative HPLC (65:35 MeOH/H_2_O, 4 mL/min) to give **11** (14.6 mg, *t*_R_ 6.7 min). Fr.2 (0.18 g) was purified by semipreparative HPLC (70:30 MeOH/H_2_O, 4 mL/min) to give **12** (48.6 mg, *t*_R_ 7.3 min). The racemic-**2** was resolved as the optically pure (+)-**2** (1.0 mg, *t*_R_ 13.7 min) and (–)-**2** (1.1 mg, *t*_R_ 10.5 min) by semi-preparative HPLC (25:75 MeCN/H_2_O, 0.9 mL/min) equipped with a chiral column (Lux^®^ 5 µm Cellulose-2, LC Column 250 × 4.6 mm, Phenomenex). Using the same procedure, the racemic **3** and **4** were chirally separated as the optically pure (-)-**3** (1.3 mg, *t*_R_ 6.5 min), (+)-**3** (1.5 mg, *t*_R_ 5.7 min), (+)-**4** (2.1 mg, *t*_R_ 4.3 min) and (–)-**4** (2.2 mg, *t*_R_ 4.8 min) by a chiral HPLC column (80:20 MeOH/H_2_O, 0.9 mL/min) (+)-**5** (35.6 mg, *t*_R_ 7.1 min) and (-)-**5** (20.9 mg, *t*_R_ 9.8 min) were chirally separated by semi-preparative HPLC (80:20 MeOH/H_2_O, 0.9 mL/min) equipped with a chiral column (Lux^®^ 5 µm i-Cellulose-5, LC Column 250 × 4.6 mm, Phenomenex).

3-[6-(2-Methylpropyl)-2-oxo-1*H*-pyrazin-3-yl]propanamide (**1**): a brown amorphous powder, mp 141–143 °C; UV (MeOH) λ_max_ (log *ε*) 217 (3.75), 230 (3.79), 262 (3.15), 325 (3.66) nm; IR (KBr) ν_max_ 3400, 3198, 2959, 2929, 2861, 1655, 1614, 1529, 1408, 1168, 575 cm^−1^; ^1^H and ^13^C NMR data (Table 1); HRESIMS *m*/*z* 246.1205 [M + Na]^+^ (calculated for C_11_H_17_O_2_N_3_Na, 246.1213).

(±)-Brevianamide X (**2**): light yellow crystal; UV (MeOH) λ_max_ (log *ε*) 233 (4.34), 245 (4.33), 287 (4.18), 329 (4.19), 357 (4.20) nm; IR (KBr) ν_max_ 3351, 2968, 1684, 1620, 1427, 1326, 1244, 1107, 1062, 1022, 979, 960, 921, 749, 655, 584, 518 cm^−1^; ^1^H and ^13^C NMR data (Table 1); ESI-MS *m*/*z* 404 [M + Na]^+^; HRESIMS *m*/*z* 404.1570 [M + Na]^+^ (calculated for C_11_H_17_O_2_N_3_Na, 404.1581). (+)-**2**: mp 241–243 °C; ${\left[\alpha \right]}_{D}^{25}$ + 44 (*c* 0.05, MeOH); ECD (0.0014 M, MeOH) *λ*_max_ (Δ*ε*) 208.5 (–6.5), 261 (+3.9), 337 (–2.5) nm. (–)-**2**: mp 220–222 °C; ${\left[\alpha \right]}_{D}^{25}$–40 (*c* 0.05, MeOH); ECD (0.001 M, MeOH) *λ*_max_ (Δ*ε*) 208.5 (+6.4), 261 (–3.8), 337 (+2.4) nm.

(±)-Brevianamide R (**3**): a colorless powder; ESI-MS *m*/*z* 402.2 [M + Na]^+^; ^1^H and ^13^C NMR (see Appendix A). (+)-**3**: mp 121–123 °C; ${\left[\alpha \right]}_{D}^{25}$ +208 (*c* 0.15, MeOH); ECD (0.007 M, MeOH) *λ*_max_ (Δ*ε*) 212 (+26.2), 258 (–19.1), 342 (+11.8) nm. (–)-**3**: mp 140–142 °C; ${\left[\alpha \right]}_{D}^{25}$ –213 (*c* 0.13, MeOH); ECD (0.007 M, MeOH) *λ*_max_ (Δ*ε*) 212 (–25.9), 258 (+16.7), 340.5 (–10.8) nm.

(±)-Brevianamide Q (**4**): a colorless powder; ESI-MS *m*/*z* 388.2 [M + Na]^+^; ^1^H and ^13^C NMR (see Appendix A). (+)-**4**: mp 141–143 °C; ${\left[\alpha \right]}_{D}^{25}$ +105 (*c* 0.2, MeOH); ECD (0.003 M, MeOH) *λ*_max_ (Δ*ε*) 212 (+21.8), 258 (–14.3), 340 (+9.3) nm. (–)-**4**: mp 160–163 °C; ${\left[\alpha \right]}_{D}^{25}$ –109 (*c* 0.2, MeOH); ECD (0.003 M, MeOH) *λ*_max_ (Δ*ε*) 213 (–19.3), 258 (+15.8), 340 (–9.7) nm.

(±)-Brevianamide V (**5**): a light yellow powder; ESI-MS *m*/*z* 372.3 [M + Na]^+^; ^1^H and ^13^C NMR (see Appendix A). (+)-**5**: mp 128–130 °C; ${\left[\alpha \right]}_{D}^{25}$ +40 (*c* 2.0, MeOH); ECD (0.0014 M, MeOH) *λ*_max_ (Δ*ε*) 212 (+24.3), 258 (–12.2), 330 (+6.3) nm. (–)-**5**: mp 131–132 °C; ${\left[\alpha \right]}_{D}^{25}$ –38 (*c* 2.0, MeOH); ECD (0.001 M, MeOH) *λ*_max_ (Δ*ε*) 213 (–22.2), 258 (+11.0), 330 (–5.7) nm.

Methyl diorcinol-4-carboxylate (**12**): a white solid; UV (MeOH) λ_max_ (log *ε*) 224 (3.42), 262 (2.95), 200 (1.36), 301 (1.23) nm; IR (KBr) ν_max_ 3401, 2954, 1656, 1598, 1454, 1322, 1262, 1207, 1161, 1104, 1031, 998, 952, 843, 677 cm^−1^; ^1^H and ^13^C NMR (see Table 2); ESI-MS *m*/*z* 311.1 [M +Na]^+^.

Methyl 3,3-di(*O*-methyl) diorcinol-4-carboxylate (**12a**): a white solid; UV (MeOH) λ_max_ (log *ε*) 222 (3.34), 228 (3.32), 279 (1.49) nm; IR (KBr) ν_max_ 3440, 2950, 1731, 1587, 1463, 1324, 1268, 1193, 1158, 1093, 1064, 1028, 996, 935, 836 cm^−1^; ^1^H and ^13^C NMR (see Table 2); HRESIMS *m*/*z* 317.1392 [M + H]^+^ (calculated for C_18_H_21_O_5_, 317.1384) (Appendix A).

### 3.5. Crystallographic Data for Racemic ***2***

Racemic **2** was obtained as a light-yellow monoclinic crystal with molecular formula C_11_H_17_O_2_N_3_ from CH_3_CH_2_OH and CH_2_Cl_2_. Space group P-1, *a* = 9.8007 (9) Å, *b* = 10.4560(12) Å, *c* = 12.9125(14) Å, *α* = 122.466(3)°, *β* = 97.4200(10)°, *γ* = 106.988(2)°, V = 1125.2(2) Å3, *Z* = 2, *D*_calcd_ = 1.259 mg/m^3^, *μ* = 0.089 mm^-1^, *F*(000) = 454, crystal size 0.34 × 0.21 × 0.13 mm, T = 298(2) K, 2.27° ≤ 2θ ≤ 25.02°, 5737 reflections collected, 3915 unique (R(int) = 0.0364). Final R indices R^1^ = 0.0310, wR^2^ = 0.0748 based on 3915 reflections with *I* > 2 *sigma*(*I*) (refinement on F^2^), 316 parameters, 0 restraint. Crystallographic data (excluding structure factors) for structure (±)-**2** in this dissertation have been deposited in the Cambridge Crystallographic Data Centre as supplementary publication number CCDC 1837585 (fax, +44-(0)-1223-336033; e-mail, deposit@ccdc.cam.ac.uk).

### 3.6. ECD Calculation Assays

The calculations were performed by using the density functional theory (DFT) using Gaussian 03. The preliminary conformational distributions search was performed by HyperChem 7.5 software. All ground-state geometries were optimized at the B3LYP/6-31G(d) level. Conformers within a 2 kcal/mol energy threshold from the global minimum were selected to calculate the electronic transitions. The overall theoretical ECD spectra were obtained according to the Boltzmann weighting of each conformers. Solvent effects of methanol solution were evaluated at the same DFT level by using the SCRF/PCM method.

### 3.7. Methylation of ***12***

To a solution of 12 (10 mg) in DMF (1 mL) were added K_2_CO_3_ (20 mg) and MeI (0.03 mL), and the mixture was then stirred for 2 h [30]. The reaction was stopped by the addition of water (10 mL) and extracted three times with EtOAc (3 × 20 mL), then EtOAc solutions were concentrated under reduced pressure. The mixture was purified by semi-HPLC (80:20 MeOH/H_2_O, 4 mL/min) to give dimethylether derivative 12a (7.4 mg, *t*_R_ 7.8 min).

### 3.8. Antimicrobial Assays

The MICs of the compounds were tested using the broth microdilution method [31,32]. The pathogenic bacteria included *B. subtilis* ATCC6051, *S. aureus* ATCC6538, *S. aureus* ATCC25923, *P. aeruginosa* ATCC10145, *E. aerogenes*, and *E. coli* ATCC11775 and the pathogenic fungi were *C. albicans* ATCC10231, and *C. glabrata* ATCC2001. Ciprofloxacin and ketoconazole were used as positive controls against bacteria and fungi, respectively. All of the microbial strains used in the bioassays are preserved at the Key Laboratory of Chemistry for Natural Products of Guizhou Province, Chinese Academy of Sciences.

### 3.9. α-Glucosidase Inhibitory Effect Assays

The inhibitory effects were assayed as described previously [12,13,33]. Absorbance of 96-well plates at 405 nm was recorded by an Epoch microplate reader (BioTek Instruments, Inc., Winooski, VT, USA). Acarbose was used as a positive control with an IC_50_ value of 255.3 μM.

## 4. Conclusions

We identified sixteen compounds from the fermentation of *Aspergillus versicolor* OUCMDZ-2738 under chemical-epigenetic modification, including a new naturally occurring diketopiperazine derivative (**1**) and two new compounds, (+)-brevianamide X ((+)-**2**) and (–)-brevianamide X ((–)-**2**). We also chirally separated the racemic brevianamides R (**3**), Q (**4**), and V (**5**) into the corresponding optically pure compounds and resolved their absolute configurations for the first time. We clearly determined the structure of the methyl diorcinol-4-carboxylate (**12**) by preparing 3,3′-dimethylether derivative (**12a**). Compounds **11** and **12** showed selective antibacterial activity against *P. aeruginosa* both with the MIC value of 17.4 and 13.9 μM, respectively.

## Figures and Tables

**Figure 1 marinedrugs-17-00006-f001:**
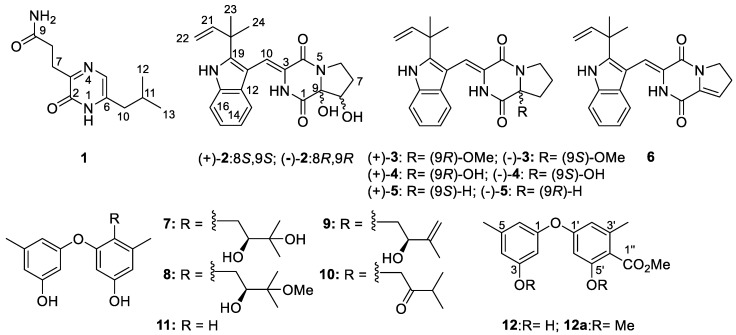
Structures of compounds **1**−**12**.

**Figure 2 marinedrugs-17-00006-f002:**
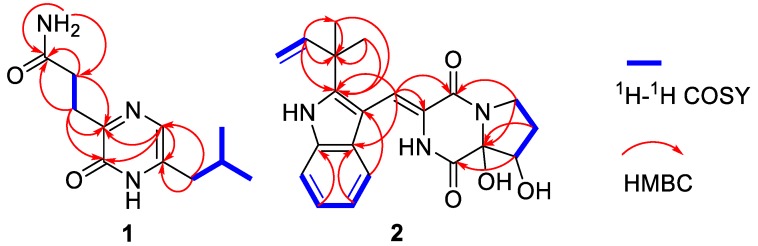
Key HMBC and ^1^H–^1^H COSY correlations of **1** and **2**.

**Figure 3 marinedrugs-17-00006-f003:**
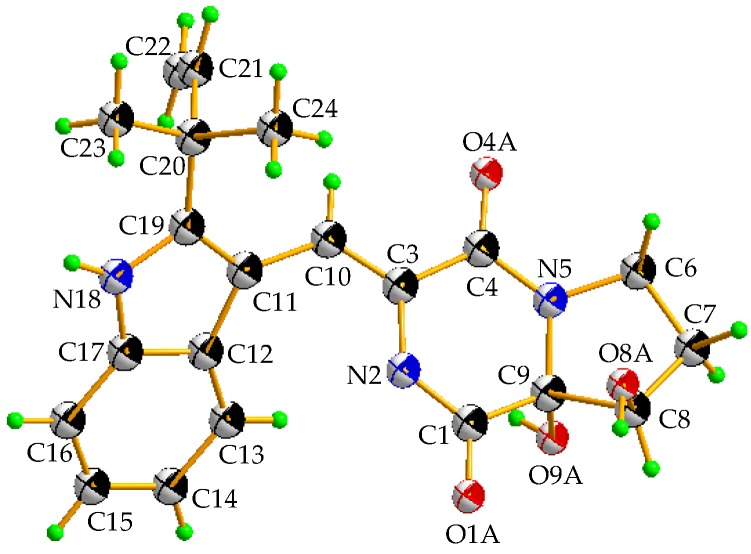
X-ray crystallographic structure of **2**.

**Figure 4 marinedrugs-17-00006-f004:**
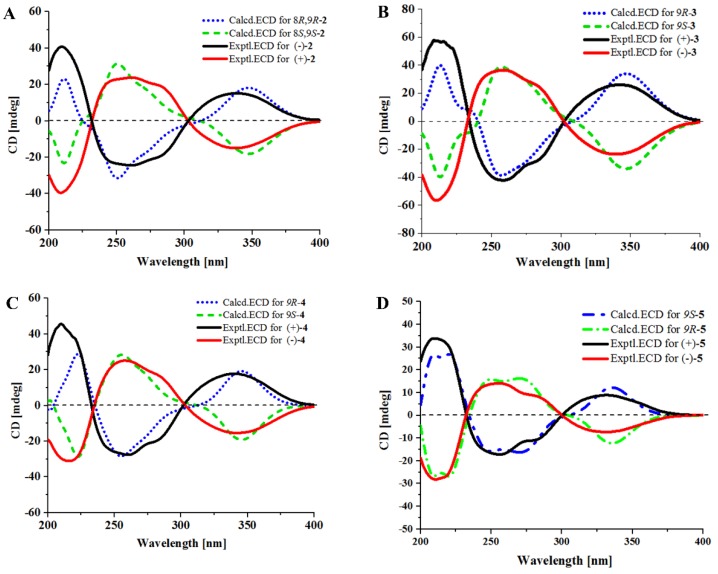
Experimental and calculated ECD spectra of **2** (**A**), **3** (**B**), **4** (**C**) and **5** (**D**).

**Figure 5 marinedrugs-17-00006-f005:**
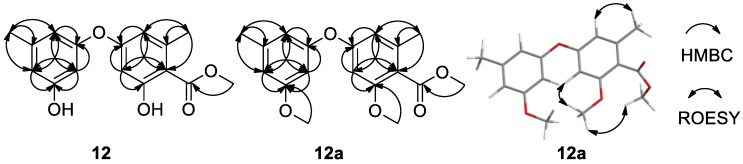
HMBC correlations of **12** and **12a**, and key ROESY correlations of **12a**.

**Figure 6 marinedrugs-17-00006-f006:**
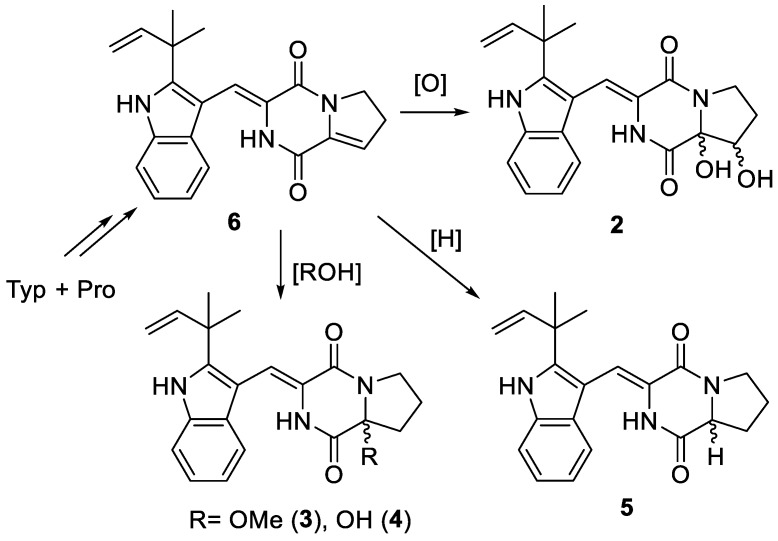
Plausible biosynthetic relationship among **2**–**6**.

**Figure 7 marinedrugs-17-00006-f007:**
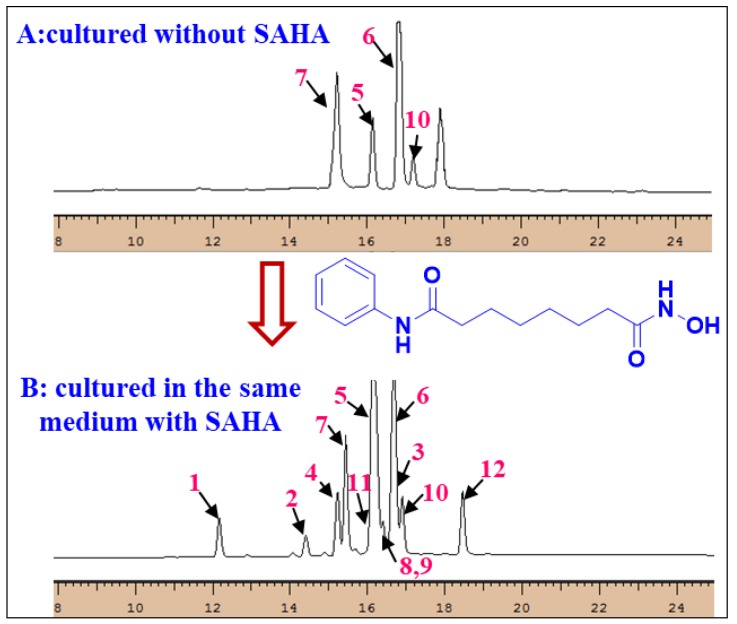
HPLC-UV profiles of the fungal metabolites with and without SAHA.

**Table 1 marinedrugs-17-00006-t001:** ^1^H (400 MHz) and ^13^C (100 MHz) NMR data of **1** in DMSO-*d*_6_ and **2** in MeOH-*d*_4_.

No.	1		2	
	δ_C_, Type	δ_H_, Mult. (*J* in Hz)	δ_C_, Type	δ_H_, Mult. (*J* in Hz)
1		12.01, brs	165.7, C	
2	156.2, C			
3	155.2, C		125.1, C	
4			161.3, C	
5	120.8, CH	7.01, s		
6	138.0, C		44.7, CH_2_	3.96, m; 3.72, m
7	27.6, CH_2_	2.79, t (7.5)	29.0, CH_2_	1.93, m; 2.41, m
8	31.1, CH_2_	2.41, t (7.5)	76.4, CH	4.41, m
9	173.7, C		91.0, C	
10	38.4, CH_2_	2.25, d (7.3)	115.1, CH	7.29, s
11	27.5, CH	1.91, m	104.5, C	
12	21.9, CH_3_	0.85, d (6.6)	127.4, C	
13	21.9, CH_3_	0.85, d (6.6)	120.2, CH	7.37, d (7.9)
14			121.3, CH	7.07, dd (7.9, 7.9)
15			122.6, CH	7.12, dd (7.9, 7.9)
16			112.6, CH	7.43, d (7.9)
17			136.8, C	
NH_2_		6.72, s; 7.30, s		
19			146.2, C	
20			40.5, C	
21			146.1, CH	6.11, dd (17.3, 10.6)
22			112.6, CH_2_	5.10, d (10.6); 5.13, d (17.3)
23			28.1, CH_3_	1.57, s
24			28.3, CH_3_	1.54, s

**Table 2 marinedrugs-17-00006-t002:** ^1^H (400 MHz) and ^13^C (100 MHz) NMR data of **12** and **12a** in DMSO-*d*_6_.

No.	12		12a		12 [27]	
	δ_C_, Type	δ_H_, Mult. (*J* in Hz)	δ_C_, Type	δ_H_, Mult. (*J* in Hz)	δ_C_, Type	δ_H_, Mult. (*J* in Hz)
1	156.4, C		156.9, C		155.9, C	
2	103.8, CH	6.22, dd (1.2, 1.2)	102.3, CH	6.42, d (1.2)	105.0, CH	6.44, brs
3	158.6, C		160.5, C		156.8, C	
4	112.1, CH	6.41, brs	110.4, CH	6.58, brs	112.6, CH	6.49, brs
5	140.5, C		140.6, C		141.2, C	
6	110.8, CH	6.30, brs	111.7, CH	6.42, d (1.2)	113.4, CH	6.37, brs
1′	159.1, C		158.4, C		162.3, C	
2′	110.9, CH	6.35, d (2.1)	111.2, CH	6.40, d (2.1)	113.0, CH	6.37, d (2.5)
3′	139.1, C		137.5, C		143.6, C	
4′	114.8, C		118.6, C		107.0, CH	
5′	157.7, C		157.6, C		165.0, C	
6′	102.7, CH	6.27, d (2.1)	100.2, CH	6.61, d (2.1)	103.2, CH	6.34, d (2.5)
1′′	168.6, C		167.6, C		172.0, C	
5-Me	21.1, CH_3_	2.20, s	21.2, CH_3_	2.25, s	21.4, CH_3_	2.28, s
3′-Me	20.2, CH_3_	2.21, s	19.0, CH_3_	2.14, s	24.2, CH_3_	2.50, s
3-OMe			55.3, CH_3_	3.72, s		
5′-OMe			56.0, CH_3_	3.72, s		
1′′-OMe	51.9, CH_3_	3.78, s	52.1, CH_3_	3.78, s	51.9, CH_3_	3.94, s
3-OH		10.21, brs				11.66, s
5′-OH		9.61, brs				

**Table 3 marinedrugs-17-00006-t003:** Antimicrobial bioassay results of active compounds.

Compounds	MIC (μM)							
	*B. subtilis*	*P. aeruginosa*	*C. perfringens*	*S. aureus* ^a^	*E. coli*	*S. aureus* ^b^	*C. albicans*	*C. glabrata*
**1**	>200	>200	>200	>200	>200	>200	>200	>200
**2**	>200	>200	>200	>200	>200	>200	>200	>200
**3**	>200	>200	>200	>200	>200	>200	>200	>200
**4**	>200	>200	>200	>200	>200	>200	>200	>200
**5**	>200	>200	>200	>200	>200	>200	>200	>200
**6**	>200	92.2	>200	184.4	>200	184.4	>200	>200
**7**	>200	>200	>200	>200	>200	>200	>200	>200
**8**	>200	46.2	>200	>200	>200	184.8	>200	>200
**9**	>200	101.9	>200	>200	>200	>200	>200	>200
**10**	>200	50.9	>200	>200	>200	>200	>200	>200
**11**	69.6	17.4	139.2	>200	>200	>200	>200	>200
**12**	>128	13.9	55.6	>200	>200	55.6	111.2	27.8
Ciprofloxacin	48.4	96.8	0.75	0.75	12.1	0.75	ND	ND
Ketoconazole	ND	ND	ND	ND	ND	ND	7.6	3.8

^a^ ATCC 6538, ^b^ ATCC 25923, ND: not detection.

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
