# Peer review of "Diketopiperazine and Diphenylether Derivatives from Marine Algae-Derived Aspergillus versicolor OUCMDZ-2738 by Epigenetic Activation"

_marinedrugs, 2018, doi:10.3390/md17010006_

Round 1
Reviewer 1 Report
The work presented by Zhu and Sun et al, describe modify the algicolous Aspericillus using chemical epigenetics with SAHA in order to increase the biodiversity of the natural product repertoire. The authors are able to purify 12 previously known compounds (compound 1 is available from commercial sources. They perform antimicrobial assay on all 12 compounds and find compounds 11 and 12 have comparable antimicrobial activity to ciprofloxacin and ketoconazole. They elucidate the structures using 1- and 2D NMR for all compounds. They also calculate ECD for compounds 2-5 and determine the X-ray crystal structure for compound 2. This work is an extension of the work that was performed and published earlier this year (Sun et al 2018). Based on the provided data and that most of the effort in cultivation, epigenetic modification, extraction and fractionation was described in a previous publication. I see the findings in this work not significant enough for the publication of marine drugs but could be published elsewhere in a lower impact journal.
Author Response
Dear editor and reviewer 1,
Thanks for your constructive comments for our manuscript. Indeed, epigenetic modification method is not an original innovation, and our structural novelty is also not good. However, we applied epigenetic modification to a new fungus strain Aspergillus versicolor OUCMDZ-2738, transforming it from a “non-genius strain” to a “genius strain”, improve the utilization and application of this strain. In addition, we chirally separated 4 pairs of racemic into the corresponding optically-pure compounds and resolved their absolute configurations for the first time. Clearly determined the structure of methyl diorcinol-4-carboxylate (12) by preparing 3,3'-dimethylether derivative (12a). These efforts can provide reference for the full development and utilization of microbial resources, the chiral separation of enantiomers and the determination of the absolute configuration of racemates. Therefore, we earnestly request you to consider allow our work published in this influential magazine.

Reviewer 2 Report
The authors report on the structural characterization and biological activities of new and known metabolites from a marine algae Aspergillus by epigenetic activation. For compound 2 reported as an enantiomeric mixture from where the enantiomer were separated; a plausible origin as artifact must be indicated, in addition its presence in the crude extract should be verified by LCMS analysis.Also for the known metabolites isolated in racemic form, a discussion must be introduced. The biological activities must be reported with concentration values in micromolar, so that to have an easier comparison among molecules with different molecular mass.
Minor comments: paragraph 3.9 is more properly moved before paragraph on biological assays.
For reference 20, remove the italic form used in the title.

Author Response
Reviewer 2
The authors report on the structural characterization and biological activities of new and known metabolites from a marine algae Aspergillus by epigenetic activation. For compound 2 reported as an enantiomeric mixture from where the enantiomer were separated;
(1) A plausible origin as artifact must be indicated, in addition its presence in the crude extract should be verified by LCMS analysis. Also for the known metabolites isolated in racemic form, a discussion must be introduced.
(2) The biological activities must be reported with concentration values in micromolar, so that to have an easier comparison among molecules with different molecular mass.
(3) Minor comments: paragraph 3.9 is more properly moved before paragraph on biological assays.
(4) For reference 20, remove the italic form used in the title.
Response:
Dear editor and reviewer 2,
Thanks for your constructive comments for our manuscript.
(1) The racemic compounds (2-5) were existed in the fermentation extract, we have added to supplementary material (see Figure S30). Therefore, they are unlikely to be artifacts. It has been reported in previous literature [20, 21] that compounds 2-5 were racemates, we chirally separated them for the first time and determined the absolute configuration through ECD calculations. These compounds may be formed by non-enzymatic catalysis, positions 8 and 9 in 6 has no steric hindrance when performing electrophilic addition or oxidation. Therefore, it is easy to form enantiomeric mixtures or excess of 2–5 (Figure 6, see paragraph 1 in page 6).
(2) We have revised the units used in the biological activities as “μM”, please see the revised manuscript.
(3) The paragraph 3.9 has been moved to 3.7, please see the revised manuscript.
(4) The format of reference 20 has been corrected, please see the revised manuscript.

Reviewer 3 Report
The manuscript is well written and well referenced. I made just a few comments on the pages of the manuscript. From your numbering of compound 1, I do not see any oxo-in number 3. Check the nomenclature of the compound. Also the supplementary figures S2 & S9 needs to be expanded to be able to see some of the splittings. In figure S17, which of the conformers is suitable for compound 2-5 or do they exist in all the conformers?

Author Response
Reviewer 3
The manuscript is well written and well referenced. I made just a few comments on the pages of the manuscript.
(1) From your numbering of compound 1, I do not see any oxo-in number 3. Check the nomenclature of the compound.
(2) Also the supplementary figures S2 & S9 needs to be expanded to be able to see some of the splittings.
(3) In figure S17, which of the conformers is suitable for compound 2-5 or do they exist in all the conformers?
Response:
Dear editor and reviewer 3,
Thanks for your constructive comments for our manuscript.
(1) We have revised the name of compoud 1 and adjusted the atomic number for the pyrazin-2(1H)-one nucleus, please see the revised manuscript.
(2) We have already expanded the supplementary figures S2 & S9, please see the revised supplementary material.
(3) Compounds 2-5 exist all the conformers in figure S17. As we described in 3.6 ECD Calculation Assays (page 9), the overall theoretical ECD spectra were obtained according to the Boltzmann weighting of each conformers.

Reviewer 4 Report
Please find my comments about the results and their presentation below:
Table 1. NMR of Compound 1: Please, look again at the interpretation of the spectrum. I have a few comments:
· C -6: Due to the construction of the compound, it is impossible to signal a triplet from one of the protons of this carbon.
· C14: There is triplet and there should be dd.
· C15: There is quartet and there should be dd.
· And also please check the coupling constants. They should be the same from protons C-15 and C-16.
· C21: there should not be dt.
Please correct sentence in lines 122-125. This sentence is incomprehensible.
Table 2. compound 12: Signals of protons at carbons C-2, C-2 ', C-6' should be singlets.
Table 2. compound 12a: Signals of protons at carbons C-2, C-2 ',C-6, C-6' should be singlets.
Please add the program of HPLC analysis.
Literature nr 7. Please correct the formatting .
Supplementary data:
Figure S2: Please add the signal to the spectral description at 12.01 ppm.
Please add to the supplementary data spectrum of compound 12.
NMR description of compounds 3, 4, 5: The authors must corrected descriptions of protons H-22 and H21. These protons are coupled with each other, so their constants should have the same values.
Author Response
Reviewer 4 Please find my comments about the results and their presentation below: Table 1. NMR of Compound 2: Please, look again at the interpretation of the spectrum. I have a few comments: (1) C-6: Due to the construction of the compound, it is impossible to signal a triplet from one of the protons of this carbon. (2) C14: There is triplet and there should be dd. C15: There is quartet and there should be dd. And also please check the coupling constants. They should be the same from protons C-15 and C-16. (3) C21: there should not be dt. (4) Please correct sentence in lines 122-125. This sentence is incomprehensible. (5) Table 2. compound 12: Signals of protons at carbons C-2, C-2', C-6' should be singlets. Table 2. compound 12a: Signals of protons at carbons C-2, C-2',C-6, C-6' should be singlets. (6) Please add the program of HPLC analysis. (7) Literature nr 7. Please correct the formatting. Supplementary data: (8) Figure S2: Please add the signal to the spectral description at 12.01 ppm. (9) Please add to the supplementary data spectrum of compound 12. (10) NMR description of compounds 3, 4, 5: The authors must corrected descriptions of protons H-22 and H21. These protons are coupled with each other, so their constants should have the same values. Response: Dear editor and reviewer 1, Thanks for your constructive comments for our manuscript. (1) We checked 1H NMR spectrum of compound 2, H-6 should be “ddd”, however, their coupling splits are not clear enough and very difficult to distinguish, therefore, they were revised to “m”, please see Table 1. (2) The triplet of H-14 and H-15 were revised to “dd”, and the coupling constants of were the same with their mutual coupling protons, please see Table 1. (3) The “dt” of H-21 was revised to “dd”, please see Table 1. (4) This sentence was revised, please see the revised manuscript. (5) In Table 2. compounds 12 and 12a: signals of protons at carbons C-2, C-2', C-6, C-6' should be singlets. However, in our 1H NMR spectrum, they present a relatively common weak coupling in the benzene ring meta-proton. Due to equipment or other reasons, H-4 and H-6 in 12, H-4 in 12a not splits, but presents as “brs”. Therefore, we didn’t revise them to “singlets”, please see Table 2 and Figure S19, S25. (6) We have added the program of HPLC analysis, please see the revised supplementary material. (7) The format of reference 7 has been corrected, please see the revised manuscript. (8) The signal 12.01 ppm in the spectral Figure S2 was added as N1-H, please see Table 1. (9) The spectrum of compounds 12 and 12a was added in supplementary material, please see the revised supplementary material Figure S19- S29. (10) We have revised compounds 3, 4, 5 accordingly, please see the revised supplementary material.

Round 2
Reviewer 1 Report
The edits made in this revised version, although makes the manuscript publishable, does increase the impact of this work to make it suitable for the publication in marine drugs. If the authors detect a new compound, not just a new-natural, this would make it suitable for publication in marine drugs
Author Response
Thanks for your suggestions. Among compounds 1-4, 8-9 and 11-12 solely produced under SAHA condition, (+)-brevianamide X [(+)-2] and (‒)-brevianamide X [(‒)-2] are two new compounds. However, compound 1 was recorded in the Aurora Fine Chemicals with ID number K13.052.726 without any physical and chemical properties. So, compound 1 was a new naturally-occurring compound. That is, we identified two new compounds and a new naturally-occurring compound from the culture under SAHA.

Reviewer 2 Report
In their revised version of the manuscript, the authors have introduced each detail related to the comments.
Author Response
Thank you for your recognition.
Reviewer 4 Report
I have no more questions for authors
Author Response
Thank you for your recognition.